# Reshaped Wirtinger Flow for Solving Quadratic System of Equations

**Huishuai Zhang**
Department of EECS
Syracuse University
Syracuse, NY 13244
hzhan23@syr.edu

**Yingbin Liang**
Department of EECS
Syracuse University
Syracuse, NY 13244
yliang06@syr.edu

## Abstract

We study the problem of recovering a vector $\boldsymbol{x} \in \mathbb{R}^n$ from its magnitude measurements $y_i = |\langle \boldsymbol{a}_i, \boldsymbol{x} \rangle|, i = 1, ..., m$. Our work is along the line of the Wirtinger flow (WF) approach Candès et al. [2015], which solves the problem by minimizing a nonconvex loss function via a gradient algorithm and can be shown to converge to a global optimal point under good initialization. In contrast to the smooth loss function used in WF, we adopt a nonsmooth but lower-order loss function, and design a gradient-like algorithm (referred to as reshaped-WF). We show that for random Gaussian measurements, reshaped-WF enjoys geometric convergence to a global optimal point as long as the number $m$ of measurements is at the order of $\mathcal{O}(n)$, where $n$ is the dimension of the unknown $\boldsymbol{x}$. This improves the sample complexity of WF, and achieves the same sample complexity as truncated-WF Chen and Candes [2015] but without truncation at gradient step. Furthermore, reshaped-WF costs less computationally than WF, and runs faster numerically than both WF and truncated-WF. Bypassing higher-order variables in the loss function and truncations in the gradient loop, analysis of reshaped-WF is simplified.

## 1 Introduction

Recovering a signal via a quadratic system of equations has gained intensive attention recently. More specifically, suppose a signal of interest $\boldsymbol{x} \in \mathbb{R}^n/\mathbb{C}^n$ is measured via random design vectors $\boldsymbol{a}_i \in \mathbb{R}^n/\mathbb{C}^n$ with the measurements $y_i$ given by

$$y_i = |\langle \boldsymbol{a}_i, \mathbf{x} \rangle|, \quad \text{for } i = 1, \cdots, m, \tag{1}$$

which can also be written equivalently in a quadratic form as $y_i' = |\langle \boldsymbol{a}_i, \mathbf{x} \rangle|^2$. The goal is to recover the signal $\boldsymbol{x}$ based on the measurements $\boldsymbol{y} = \{y_i\}_{i=1}^m$ and the design vectors $\{\boldsymbol{a}_i\}_{i=1}^m$. Such a problem arises naturally in the phase retrieval applications, in which the sign/phase of the signal is to be recovered from only measurements of magnitudes. Various algorithms have been proposed to solve this problem since 1970s. The error-reduction methods proposed in Gerchberg [1972], Fienup [1982] work well empirically but lack theoretical guarantees. More recently, convex relaxation of the problem has been formulated, for example, via phase lifting Chai et al. [2011], Candès et al. [2013], Gross et al. [2015] and via phase cut Waldspurger et al. [2015], and the correspondingly developed algorithms typically come with performance guarantee. The reader can refer to the review paper Shechtman et al. [2015] to learn more about applications and algorithms of the phase retrieval problem.

While with good theoretical guarantee, these convex methods often suffer from computational complexity particularly when the signal dimension is large. On the other hand, more efficient nonconvex approaches have been proposed and shown to recover the true signal as long as initialization is good enough. Netrapalli et al. [2013] proposed *AltMinPhase* algorithm, which alternatively updates the

phase and the signal with each signal update solving a least-squares problem, and showed that Alt-MinPhase converges linearly and recovers the true signal with $\mathcal{O}(n \log^3 n)$ Gaussian measurements. More recently, Candès et al. [2015] introduces *Wirtinger flow* (WF) algorithm, which guarantees signal recovery via a simple gradient algorithm with only $\mathcal{O}(n \log n)$ Gaussian measurements and attains $\epsilon-$accuracy within $\mathcal{O}(mn^2 \log 1/\epsilon)$ flops. More specifically, WF obtains good initialization by the spectral method, and then minimizes the following nonconvex loss function

$$\ell_{WF}(\boldsymbol{z}) := \frac{1}{4m} \sum_{i=1}^{m} (|\boldsymbol{a}_i^T \boldsymbol{z}|^2 - y_i^2)^2, \tag{2}$$

via the gradient descent scheme.

WF was further improved by *truncated Wirtinger flow* (truncated-WF) algorithm proposed in Chen and Candes [2015], which adopts a Poisson loss function of $|\boldsymbol{a}_i^T \boldsymbol{z}|^2$, and keeps only well-behaved measurements based on carefully designed truncation thresholds for calculating the initial seed and every step of gradient . Such truncation assists to yield linear convergence with certain fixed step size and reduces both the sample complexity to $(\mathcal{O}(n))$ and the convergence time to $(\mathcal{O}(mn \log 1/\epsilon))$.

It can be observed that WF uses the quadratic loss of $|\boldsymbol{a}_i^T \boldsymbol{z}|^2$ so that the optimization objective is a *smooth* function of $\boldsymbol{a}_i^T \boldsymbol{z}$ and the gradient step becomes simple. But this comes with a cost of a quartic loss function. In this paper, we adopt the quadratic loss of $|\boldsymbol{a}_i^T \boldsymbol{z}|$. Although the loss function is not smooth everywhere, it reduces the order of $\boldsymbol{a}_i^T \boldsymbol{z}$ to be two, and the general curvature can be more amenable to convergence of the gradient algorithm. The goal of this paper is to explore potential advantages of such a nonsmooth lower-order loss function.

## 1.1   Our Contribution

This paper adopts the following loss function[1]

$$\ell(\boldsymbol{z}) := \frac{1}{2m} \sum_{i=1}^{m} \left( |\boldsymbol{a}_i^T \boldsymbol{z}| - y_i \right)^2. \tag{3}$$

Compared to the loss function (2) in WF that adopts $|\boldsymbol{a}_i^T \boldsymbol{z}|^2$, the above loss function adopts the absolute value/magnitude $|\boldsymbol{a}_i^T \boldsymbol{z}|$ and hence has lower-order variables. For such a nonconvex and nonsmooth loss function, we develop a gradient descent-like algorithm, which sets zero for the "gradient" component corresponding to nonsmooth samples. We refer to such an algorithm together with truncated initialization using spectral method as *reshaped Wirtinger flow* (reshaped-WF). We show that the lower-order loss function has great advantage in both statistical and computational efficiency, although scarifying smoothness. In fact, the curvature of such a loss function behaves similarly to that of a least-squares loss function in the neighborhood of global optimums (see Section 2.2), and hence reshaped-WF converges fast. The nonsmoothness does not significantly affect the convergence of the algorithm because only with negligible probability the algorithm encounters nonsmooth points for some samples, which furthermore are set not to contribute to the gradient direction by the algorithm. We summarize our main results as follows.

- Statistically, we show that reshaped-WF recovers the true signal with $\mathcal{O}(n)$ samples, when the design vectors consist of *independently and identically distributed* (i.i.d.) Gaussian entries, which is optimal in the order sense. Thus, even *without truncation* in gradient steps (truncation only in initialization stage), reshaped WF improves the sample complexity $\mathcal{O}(n \log n)$ of WF, and achieves the same sample complexity as truncated-WF. It is thus more robust to random measurements.

- Computationally, reshaped-WF converges geometrically, requiring $\mathcal{O}(mn \log 1/\epsilon)$ flops to reach $\epsilon-$accuracy. Again, without truncation in gradient steps, reshaped-WF improves computational cost $\mathcal{O}(mn^2 \log(1/\epsilon))$ of WF and achieves the same computational cost as truncated-WF. Numerically, reshaped-WF is generally two times faster than truncated-WF and four to six times faster than WF in terms of the number of iterations and time cost.

Compared to WF and truncated-WF, our technical proof of performance guarantee is much simpler, because the lower-order loss function allows to bypass higher-order moments of variables and

truncation in gradient steps. We also anticipate that such analysis is more easily extendable. On the other hand, the new form of the gradient step due to nonsmoothness of absolute function requires new developments of bounding techniques.

## 1.2 Connection to Related Work

Along the line of developing nonconvex algorithms with global performance guarantee for the phase retrieval problem, Netrapalli et al. [2013] developed alternating minimization algorithm, Candès et al. [2015], Chen and Candes [2015], Zhang et al. [2016], Cai et al. [2015] developed/studied first-order gradient-like algorithms, and a recent study Sun et al. [2016] characterized geometric structure of the nonconvex objective and designed a second-order trust-region algorithm. Also notably is Wei [2015], which empirically demonstrated fast convergence of a so-called Kaczmarz stochastic algorithm. This paper is most closely related to Candès et al. [2015], Chen and Candes [2015], Zhang et al. [2016], but develops a new gradient-like algorithm based on a lower-order nonsmooth (as well as nonconvex) loss function that yields advantageous statistical/computational efficiency.

Various algorithms have been proposed for minimizing a general nonconvex nonsmooth objective, such as gradient sampling algorithm Burke et al. [2005], Kiwiel [2007] and majorization-minimization method Ochs et al. [2015]. These algorithms were often shown to convergence to critical points which may be local minimizers or saddle points, without explicit characterization of convergence rate. In contrast, our algorithm is specifically designed for the phase retrieval problem, and can be shown to converge linearly to global optimum under appropriate initialization.

The advantage of nonsmooth loss function exhibiting in our study is analogous in spirit to that of the rectifier activation function (of the form $\max\{0, \cdot\}$) in neural networks. It has been shown that rectified linear unit (*ReLU*) enjoys superb advantage in reducing the training time Krizhevsky et al. [2012] and promoting sparsity Glorot et al. [2011] over its counterparts of sigmoid and hyperbolic tangent functions, in spite of non-linearity and non-differentiability at zero. Our result in fact also demonstrates that a nonsmooth but simpler loss function yields improved performance.

## 1.3 Paper Organization and Notations

The rest of this paper is organized as follows. Section 2 describes reshaped-WF algorithm in detail and establishes its performance guarantee. In particular, Section 2.2 provides intuition about why reshaped-WF is fast. Section 3 compares reshaped-WF with other competitive algorithms numerically. Finally, Section 4 concludes the paper with comments on future directions.

Throughout the paper, boldface lowercase letters such as $\boldsymbol{a}_i, \boldsymbol{x}, \boldsymbol{z}$ denote vectors, and boldface capital letters such as $\boldsymbol{A}, \boldsymbol{Y}$ denote matrices. For two matrices, $\boldsymbol{A} \preceq \boldsymbol{B}$ means that $\boldsymbol{B} - \boldsymbol{A}$ is positive definite. The indicator function $\mathbf{1}_A = 1$ if the event $A$ is true, and $\mathbf{1}_A = 0$ otherwise. The Euclidean distance between two vectors up to a global sign difference is defined as $\text{dist}(\boldsymbol{z}, \boldsymbol{x}) := \min\{\|\boldsymbol{z} - \boldsymbol{x}\|, \|\boldsymbol{z} + \boldsymbol{x}\|\}$.

## 2 Algorithm and Performance Guarantee

In this paper, we wish to recover a signal $\boldsymbol{x} \in \mathbb{R}^n$ based on $m$ measurements $y_i$ given by

$$y_i = |\langle \boldsymbol{a}_i, \mathbf{x} \rangle|, \quad \text{for } i = 1, \cdots, m, \tag{4}$$

where $\boldsymbol{a}_i \in \mathbb{R}^n$ for $i = 1, \cdots, m$ are known measurement vectors generated by Gaussian distribution $\mathcal{N}(0, \boldsymbol{I}_{n \times n})$. We focus on the real-valued case in analysis, but the algorithm designed below is applicable to the complex-valued case and the case with *coded diffraction pattern* (CDP) as we demonstrate via numerical results in Section 3.

We design reshaped-WF (see Algorithm 1) for solving the above problem, which contains two stages: spectral initialization and gradient loop. Suggested values for parameters are $\alpha_l = 1, \alpha_u = 5$ and $\mu = 0.8$. The scaling parameter in $\lambda_0$ and the conjugate transpose $\boldsymbol{a}_i^*$ allow the algorithm readily applicable to complex and CDP cases. We next describe the two stages of the algorithm in detail in Sections 2.1 and 2.2, respectively, and establish the convergence of the algorithm in Section 2.3.

## 2.1 Initialization via Spectral Method

We first note that initialization can adopt the spectral initialization method for WF in Candès et al. [2015] or that for truncated-WF in Chen and Candes [2015], both of which are based on $|\boldsymbol{a}_i^* \boldsymbol{x}|^2$. Here, we propose an alternative initialization in Algorithm 1 that uses magnitude $|\boldsymbol{a}_i^* \boldsymbol{x}|$ instead, and truncates samples with both lower and upper thresholds as in (5). We show that such initialization achieves smaller sample complexity than WF and the same order-level sample complexity as truncated-WF, and furthermore, performs better than both WF and truncated-WF numerically.

---

**Algorithm 1** Reshaped Wirtinger Flow

---

**Input**: $\boldsymbol{y} = \{y_i\}_{i=1}^m, \{\boldsymbol{a}_i\}_{i=1}^m$;

**Parameters:** Lower and upper thresholds $\alpha_l, \alpha_u$ for truncation in initialization, stepsize $\mu$;

**Initialization**: Let $\boldsymbol{z}^{(0)} = \lambda_0 \tilde{\boldsymbol{z}}$, where $\lambda_0 = \frac{mn}{\sum_{i=1}^m \|\boldsymbol{a}_i\|_1} \cdot \left( \frac{1}{m} \sum_{i=1}^m y_i \right)$ and $\tilde{\boldsymbol{z}}$ is the leading eigenvector of

$$\boldsymbol{Y} := \frac{1}{m} \sum_{i=1}^m y_i \boldsymbol{a}_i \boldsymbol{a}_i^* \mathbf{1}_{\{\alpha_l \lambda_0 < y_i < \alpha_u \lambda_0\}}. \tag{5}$$

**Gradient loop**: for $t = 0 : T - 1$ do

$$\boldsymbol{z}^{(t+1)} = \boldsymbol{z}^{(t)} - \frac{\mu}{m} \sum_{i=1}^m \left( \boldsymbol{a}_i^* \boldsymbol{z}^{(t)} - y_i \cdot \frac{\boldsymbol{a}_i^* \boldsymbol{z}^{(t)}}{|\boldsymbol{a}_i^* \boldsymbol{z}^{(t)}|} \right) \boldsymbol{a}_i. \tag{6}$$

**Output** $\boldsymbol{z}^{(T)}$.

---

Our initialization consists of estimation of both the norm and direction of $\boldsymbol{x}$. The norm estimation of $\boldsymbol{x}$ is given by $\lambda_0$ in Algorithm 1 with mathematical justification in Suppl. A. Intuitively, with real Gaussian measurements, the scaling coefficient $\frac{mn}{\sum_{i=1}^m \|\boldsymbol{a}_i\|_1} \approx \sqrt{\frac{\pi}{2}}$. Moreover, $y_i = |\boldsymbol{a}_i^T \boldsymbol{x}|$ are independent sub-Gaussian random variables for $i = 1, \ldots, m$ with mean $\sqrt{\frac{2}{\pi}} \|\boldsymbol{x}\|$, and thus $\frac{1}{m} \sum_{i=1}^m y_i \approx \sqrt{\frac{2}{\pi}} \|\boldsymbol{x}\|$. Combining these two facts yields the desired argument.

The direction of $\boldsymbol{x}$ is approximated by the leading eigenvector of $\boldsymbol{Y}$, because $\boldsymbol{Y}$ approaches $\mathrm{E}[\boldsymbol{Y}]$ by concentration of measure and the leading eigenvector of $\mathrm{E}[\boldsymbol{Y}]$ takes the form $c\boldsymbol{x}$ for some scalar $c \in \mathbb{R}$. We note that (5) involves truncation of samples from both sides, in contrast to truncation only by an upper threshold in Chen and Candes [2015]. This is because $y_i = |\boldsymbol{a}_i^T \boldsymbol{x}|^2$ in Chen and Candes [2015] so that small $|\boldsymbol{a}_i^T \boldsymbol{x}|$ is further reduced by the square power to contribute less in $\boldsymbol{Y}$, but small values of $y_i = |\boldsymbol{a}_i^T \boldsymbol{x}|$ can still introduce considerable contributions and hence should be truncated by the lower threshold.

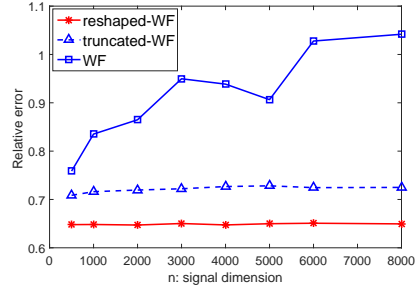

We next provide the formal statement of the performance guarantee for the initialization step that we propose. The proof adapts that in Chen and Candes [2015] and is provided in Suppl. A.

Figure 1: Comparison of three initialization methods with $m = 6n$ and 50 iterations using power method.

**Proposition 1.** *Fix $\delta > 0$. The initialization step in Algorithm 1 yields $\boldsymbol{z}^{(0)}$ satisfying $\|\boldsymbol{z}^{(0)} - \boldsymbol{x}\| \le \delta \|\boldsymbol{x}\|$ with probability at least $1 - \exp(-c'm\epsilon^2)$, if $m > C(\delta, \epsilon)n$, where $C$ is a positive number only affected by $\delta$ and $\epsilon$, and $c'$ is some positive constant.*

Finally, Figure 1 demonstrates that reshaped-WF achieves better initialization accuracy in terms of the relative error $\frac{\|\boldsymbol{z}^{(0)} - \boldsymbol{x}\|}{\|\boldsymbol{x}\|}$ than WF and truncated-WF with Gaussian measurements.

## 2.2 Gradient Loop and Why Reshaped-WF is Fast

The gradient loop of Algorithm 1 is based on the loss function (3), which is rewritten below

$$\ell(\boldsymbol{z}) := \frac{1}{2m} \sum_{i=1}^m \left( |\boldsymbol{a}_i^T \boldsymbol{z}| - y_i \right)^2. \tag{7}$$

We define the update direction as

$$\nabla \ell(\boldsymbol{z}) := \frac{1}{m} \sum_{i=1}^m \left( \boldsymbol{a}_i^T \boldsymbol{z} - y_i \cdot \mathrm{sgn}(\boldsymbol{a}_i^T \boldsymbol{z}) \right) \boldsymbol{a}_i = \frac{1}{m} \sum_{i=1}^m \left( \boldsymbol{a}_i^T \boldsymbol{z} - y_i \cdot \frac{\boldsymbol{a}_i^T \boldsymbol{z}}{|\boldsymbol{a}_i^T \boldsymbol{z}|} \right) \boldsymbol{a}_i, \tag{8}$$

where sgn$(\cdot)$ is the sign function for nonzero arguments. We further set sgn$(0) = 0$ and $\frac{0}{|0|} = 0$. In fact, $\nabla\ell(\boldsymbol{z})$ equals the gradient of the loss function (7) if $\boldsymbol{a}_i^T \boldsymbol{z} \neq 0$ for all $i = 1, ..., m$. For samples with nonsmooth point, i.e., $\boldsymbol{a}_i^T \boldsymbol{z} = 0$, we adopt Fréchet superdifferential Kruger [2003] for nonconvex function to set the corresponding gradient component to be zero (as zero is an element in Fréchet superdifferential). With abuse of terminology, we still refer to $\nabla\ell(\boldsymbol{z})$ in (8) as "gradient" for simplicity, which rather represents the update direction in the gradient loop of Algorithm 1.

We next provide the intuition about why reshaped WF is fast. Suppose that the spectral method sets an initial point in the neighborhood of ground truth $\boldsymbol{x}$. We compare reshaped-WF with the following problem of solving $\boldsymbol{x}$ from *linear equations* $y_i = \langle \boldsymbol{a}_i, \mathbf{x} \rangle$ with $y_i$ and $\boldsymbol{a}_i$ for $i = 1, \ldots, m$ given. In particular, we note that this problem has both magnitude and sign observation of the measurements. Further suppose that the least-squares loss is used and gradient descent is applied to solve this problem. Then the gradient is given by

$$\text{Least square gradient:} \quad \nabla\ell_{LS}(\boldsymbol{z}) = \frac{1}{m} \sum_{i=1}^{m} \left( \boldsymbol{a}_i^T \boldsymbol{z} - \boldsymbol{a}_i^T \boldsymbol{x} \right) \boldsymbol{a}_i. \tag{9}$$

We now argue informally that the gradient (8) of reshaped-WF behaves similarly to the least-squares gradient (9). For each $i$, the two gradient components are close if $|\boldsymbol{a}_i^T \boldsymbol{x}| \cdot$ sgn$(\boldsymbol{a}_i^T \boldsymbol{z})$ is viewed as an estimate of $\boldsymbol{a}_i^T \boldsymbol{x}$. The following lemma (see Suppl. B.2 for the proof) shows that if dist$(\boldsymbol{z}, \boldsymbol{x})$ is small (guaranteed by initialization), then $\boldsymbol{a}_i^T \boldsymbol{z}$ has the same sign with $\boldsymbol{a}_i^T \boldsymbol{x}$ for large $|\boldsymbol{a}_i^T \boldsymbol{x}|$.

**Lemma 1.** *Let $\boldsymbol{a}_i \sim \mathcal{N}(0, \boldsymbol{I}_{n \times n})$. For any given $\boldsymbol{x}$ and $\boldsymbol{z}$ satisfying $\|\boldsymbol{x} - \boldsymbol{z}\| < \frac{\sqrt{2}-1}{\sqrt{2}}\|\boldsymbol{x}\|$, we have*

$$\mathbb{P}\{(\boldsymbol{a}_i^T \boldsymbol{x})(\boldsymbol{a}_i^T \boldsymbol{z}) < 0 \big| (\boldsymbol{a}_i^T \boldsymbol{x})^2 = t\|\boldsymbol{x}\|^2\} \leq erfc\left(\frac{\sqrt{t}\|\boldsymbol{x}\|}{2\|\boldsymbol{h}\|}\right), \tag{10}$$

*where $\boldsymbol{h} = \boldsymbol{z} - \boldsymbol{x}$ and $erfc(u) := \frac{2}{\sqrt{\pi}}\int_u^\infty \exp(-\tau^2)d\tau$.*

It is easy to observe in (10) that large $\boldsymbol{a}_i^T \boldsymbol{x}$ is likely to have the same sign as $\boldsymbol{a}_i^T \boldsymbol{z}$ so that the corresponding gradient components in (8) and (9) are likely equal, whereas small $\boldsymbol{a}_i^T \boldsymbol{x}$ may have different sign as $\boldsymbol{a}_i^T \boldsymbol{z}$ but contributes less to the gradient. Hence, overall the two gradients (8) and (9) should be close to each other with a large probability.

This fact can be further verified numerically. Figure 2(a) illustrates that reshaped-WF takes almost the *same* number of iterations for recovering a signal (with only magnitude information) as the least-squares gradient descent method for recovering a signal (with both magnitude and sign information).

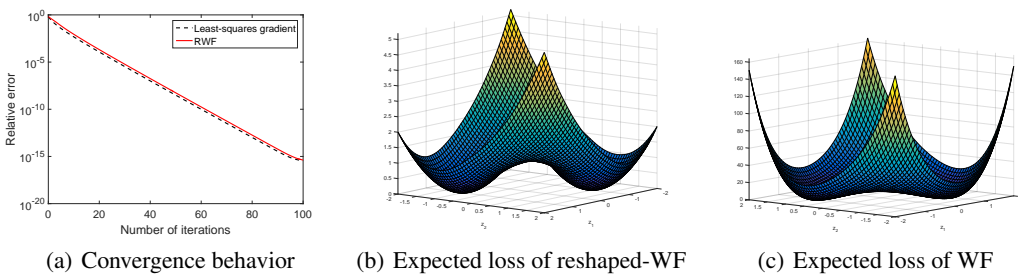

(a) Convergence behavior        (b) Expected loss of reshaped-WF        (c) Expected loss of WF

Figure 2: Intuition of why reshaped-WF fast. (a) Comparison of convergence behavior between reshaped-WF and least-squares gradient descent. Initialization and parameters are the same for two methods: $n = 1000$, $m = 6n$, step size $\mu = 0.8$. (b) Expected loss function of reshaped-WF for $\boldsymbol{x} = [1 - 1]^T$. (c) Expected loss function of WF for $\boldsymbol{x} = [1 - 1]^T$.

Figure 2(b) further illustrates that the expected loss surface of reshaped-WF (see Suppl. B for expression) behaves similarly to a quadratic surface around the global optimums as compared to the expected loss surface for WF (see Suppl. B for expression) in Figure 2(c).

## 2.3 Geometric Convergence of Reshaped-WF

We characterize the convergence of reshaped-WF in the following theorem.

**Theorem 1.** *Consider the problem of solving any given $\boldsymbol{x} \in \mathbb{R}^n$ from a system of equations (4) with Gaussian measurement vectors. There exist some universal constants $\mu_0 > 0$ ($\mu_0$ can be set as 0.8 in practice), $0 < \rho, \nu < 1$ and $c_0, c_1, c_2 > 0$ such that if $m \geq c_0 n$ and $\mu < \mu_0$, then with probability at least $1 - c_1 \exp(-c_2 m)$, Algorithm 1 yields*

$$dist(\boldsymbol{z}^{(t)}, \boldsymbol{x}) \leq \nu(1-\rho)^t \|\boldsymbol{x}\|, \quad \forall t \in \mathbb{N}. \tag{11}$$

*Outline of the Proof.* We outline the proof here with details relegated to Suppl. C. Compared to WF and truncated-WF, our proof is much simpler due to the lower-order loss function that reshaped-WF relies on.

The central idea is to show that within the neighborhood of global optimums, reshaped-WF satisfies the *Regularity Condition* $\mathsf{RC}(\mu, \lambda, c)$ Chen and Candes [2015], i.e.,

$$\langle \nabla \ell(\boldsymbol{z}), \boldsymbol{h} \rangle \geq \frac{\mu}{2} \|\nabla \ell(\boldsymbol{z})\|^2 + \frac{\lambda}{2} \|\boldsymbol{h}\|^2 \tag{12}$$

for all $\boldsymbol{z}$ and $\boldsymbol{h} = \boldsymbol{z} - \boldsymbol{x}$ obeying $\|\boldsymbol{h}\| \leq c\|\boldsymbol{x}\|$, where $0 < c < 1$ is some constant. Then, as shown in Chen and Candes [2015], once the initialization lands into this neighborhood, geometric convergence can be guaranteed, i.e.,

$$dist^2 (\boldsymbol{z} + \mu \nabla \ell(\boldsymbol{z}), \boldsymbol{x}) \leq (1 - \mu\lambda)dist^2(\boldsymbol{z}, \boldsymbol{x}), \tag{13}$$

for any $\boldsymbol{z}$ with $\|\boldsymbol{z} - \boldsymbol{x}\| \leq \epsilon\|\boldsymbol{x}\|$.

Lemmas 2 and 3 in Suppl.C yield that

$$\langle \nabla \ell(\boldsymbol{z}), \boldsymbol{h} \rangle \geq (1 - 0.26 - 2\epsilon)\|\boldsymbol{h}\|^2 = (0.74 - 2\epsilon)\|\boldsymbol{h}\|^2.$$

And Lemma 4 in Suppl.C further yields that

$$\|\nabla \ell(\boldsymbol{z})\| \leq (1 + \delta) \cdot 2\|\boldsymbol{h}\|. \tag{14}$$

Therefore, the above two bounds imply that Regularity Condition (12) holds for $\mu$ and $\lambda$ satisfying

$$0.74 - 2\epsilon \geq \frac{\mu}{2} \cdot 4(1 + \delta)^2 + \frac{\lambda}{2}. \tag{15}$$

$\square$

We note that (15) implies an upper bound $\mu \leq \frac{0.74}{2} = 0.37$, by taking $\epsilon$ and $\delta$ to be sufficiently small. This suggests a range to set the step size in Algorithm 1. However, in practice, $\mu$ can be set much larger than such a bound, say 0.8, while still keeping the algorithm convergent. This is because the coefficients in the proof are set for convenience of proof rather than being tightly chosen.

Theorem 1 indicates that reshaped-WF recovers the true signal with $\mathcal{O}(n)$ samples, which is order-level optimal. Such an algorithm improves the sample complexity $\mathcal{O}(n \log n)$ of WF. Furthermore, reshaped-WF does not require truncation of weak samples in the gradient step to achieve the same sample complexity as truncated-WF. This is mainly because reshaped-WF benefits from the lower-order loss function given in (7), the curvature of which behaves similarly to the least-squares loss function locally as we explain in Section 2.2.

Theorem 1 also suggests that reshaped-WF converges geometrically at a constant step size. To reach $\epsilon-$accuracy, it requires computational cost of $\mathcal{O}(mn \log 1/\epsilon)$ flops, which is better than WF ($\mathcal{O}(mn^2 \log(1/\epsilon))$). Furthermore, it does not require truncation in gradient steps to reach the same computational cost as truncated-WF. Numerically, as we demonstrate in Section 3, reshaped-WF is two times faster than truncated-WF and four to six times faster than WF in terms of both iteration count and time cost in various examples.

Although our focus in this paper is on the noise-free model, reshaped-WF can be applied to noisy models as well. Suppose the measurements are corrupted by bounded noises $\{\eta_i\}_{i=1}^m$ satisfying $\|\boldsymbol{\eta}\|/\sqrt{m} \leq c\|\boldsymbol{x}\|$. Then by adapting the proof of Theorem 1, it can be shown that the gradient loop of reshaped-WF is robust such that

$$\mathrm{dist}(\boldsymbol{z}^{(t)}, \boldsymbol{x}) \lesssim \frac{\|\boldsymbol{\eta}\|}{\sqrt{m}} + (1-\rho)^t\|\boldsymbol{x}\|, \quad \forall t \in \mathbb{N}, \tag{16}$$

for some $\rho \in (0, 1)$. The numerical result under the Poisson noise model in Section 3 further corroborates the stability of reshaped-WF.

Table 1: Comparison of iteration count and time cost among algorithms ($n = 1000, m = 8n$)

|  | Algorithms | reshaped-WF | truncated-WF | WF | AltMinPhase |
|---|---|---|---|---|---|
| real case | iterations | 72 | 182 | 319.2 | **5.8** |
|  | time cost(s) | **0.477** | 1.232 | 2.104 | 0.908 |
| complex case | iterations | 272.7 | 486.7 | 915.4 | **156** |
|  | time cost(s) | **6.956** | 12.815 | 23.306 | 93.22 |

## 3 Numerical Comparison with Other Algorithms

In this section, we demonstrate the numerical efficiency of reshaped-WF by comparing its performance with other competitive algorithms. Our experiments are run not only for real-valued case but also for complex-valued and CDP cases. All the experiments are implemented in Matlab 2015b and carried out on a computer equipped with Intel Core i7 3.4GHz CPU and 12GB RAM.

We first compare the sample complexity of reshaped-WF with those of truncated-WF and WF via the empirical successful recovery rate versus the number of measurements. For reshaped-WF, we follow Algorithm 1 with suggested parameters. For truncated-WF and WF, we use the codes provided in the original papers with the suggested parameters. We conduct the experiment for real, complex and CDP cases respectively. For real and complex cases, we set the signal dimension $n$ to be 1000, and the ratio $m/n$ take values from 2 to 6 by a step size 0.1. For each $m$, we run 100 trials and count the number of successful trials. For each trial, we run a fixed number of iterations $T = 1000$ for all algorithms. A trial is declared to be successful if $z^{(T)}$, the output of the algorithm, satisfies dist$(z^{(T)}, x)/\|x\| \leq 10^{-5}$. For the real case, we generate signal $x \sim \mathcal{N}(0, I_{n \times n})$, and the measurement vectors $a_i \sim \mathcal{N}(0, I_{n \times n})$ i.i.d. for $i = 1, \ldots, m$. For the complex case, we generate signal $x \sim \mathcal{N}(0, I_{n \times n}) + j\mathcal{N}(0, I_{n \times n})$ and measurements $a_i \sim \frac{1}{2}\mathcal{N}(0, I_{n \times n}) + j\frac{1}{2}\mathcal{N}(0, I_{n \times n})$ i.i.d. for $i = 1, \ldots, m$. For the CDP case, we generate signal $x \sim \mathcal{N}(0, I_{n \times n}) + j\mathcal{N}(0, I_{n \times n})$ that yields measurements

$$y^{(l)} = |FD^{(l)}x|, \quad 1 \leq l \leq L, \tag{17}$$

where $F$ represents the discrete Fourier transform (DFT) matrix, and $D^{(l)}$ is a diagonal matrix (mask). We set $n = 1024$ for convenience of FFT and $m/n = L = 1, 2, \ldots, 8$. All other settings are the same as those for the real case.

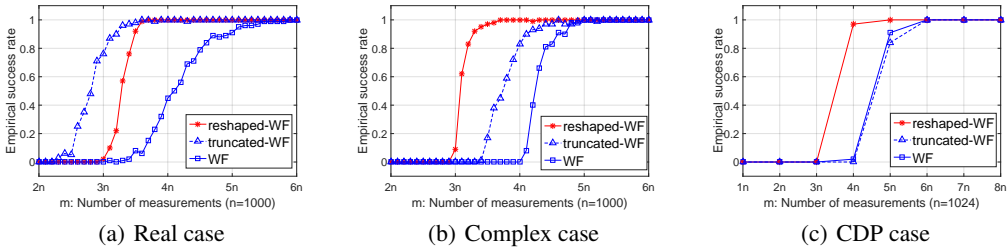

(a) Real case  (b) Complex case  (c) CDP case

Figure 3: Comparison of sample complexity among reshaped-WF, truncated-WF and WF.

Figure 3 plots the fraction of successful trials out of 100 trials for all algorithms, with respect to $m$. It can be seen that for although reshaped-WF outperforms only WF (not truncated-WF) for the real case, it outperforms both WF and truncated-WF for complex and CDP cases. An intuitive explanation for the real case is that a substantial number of samples with small $|a_i^T z|$ can deviate gradient so that truncation indeed helps to stabilize the algorithm if the number of measurements is not large. Furthermore, reshaped-WF exhibits *shaper* transition than truncated-WF and WF.

We next compare the convergence rate of reshaped-WF with those of truncated-WF, WF and AltMinPhase. We run all of the algorithms with suggested parameter settings in the original codes. We generate signal and measurements in the same way as those in the first experiment with $n = 1000, m = 8n$. All algorithms are seeded with reshaped-WF initialization. In Table 1, we list the number of iterations and time cost for those algorithms to achieve the relative error of $10^{-14}$ averaged over 10 trials. Clearly, reshaped-WF takes many fewer iterations as well as runing much faster than truncated-WF and WF. Although reshaped-WF takes more iterations than AltMinPhase, it runs much faster than

AltMinPhase due to the fact that each iteration of AltMinPhase needs to solve a least-squares problem that takes much longer time than a simple gradient update in reshaped-WF.

We also compare the performance of the above algorithms on the recovery of a real image from the Fourier intensity measurements (2D CDP with the number of masks $L = 16$). The image (provided in Suppl.D) is the Milky Way Galaxy with resolution $1920 \times 1080$. Table 2 lists the number of iterations and time cost of the above four algorithms to achieve the relative error of $10^{-15}$. It can be seen that reshaped-WF outperforms all other three algorithms in computational time cost. In particular, it is two times faster than truncated-WF and six times faster than WF in terms of both the number of iterations and computational time cost.

Table 2: Comparison of iterations and time cost among algorithms on Galaxy image ($L = 16$)

| Algorithms | reshaped-WF | truncated-WF | WF | AltMinPhase |
|---|---|---|---|---|
| iterations | **65** | 160 | 420 | 110 |
| time cost(s) | **141** | 567 | 998 | 213 |

We next demonstrate the robustness of reshaped-WF to noise corruption and compare it with truncated-WF. We consider the phase retrieval problem in imaging applications, where random Poisson noises are often used to model the sensor and electronic noise Fogel et al. [2013]. Specifically, the noisy measurements of intensity can be expressed as $\mathrm{y}_i = \sqrt{\alpha \cdot \mathrm{Poisson}\left(|\boldsymbol{a}_i^T \boldsymbol{x}|^2 / \alpha\right)}, \quad \text{for } i = 1, 2, ...m$ where $\alpha$ denotes the level of input noise, and $\mathrm{Poisson}(\lambda)$ denotes a random sample generated by the Poisson distribution with mean $\lambda$. It can be observed from Figure 4 that reshaped-WF performs better than truncated-WF in terms of recovery accuracy under different noise levels.

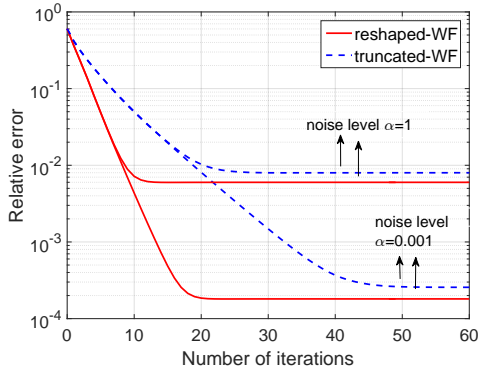

Figure 4: Comparison of relative error under Poisson noise between reshaped-WF and truncated WF.

## 4   Conclusion

In this paper, we proposed reshaped-WF to recover a signal from a quadratic system of equations, based on a *nonconvex and nonsmooth* quadratic loss function of absolute values of measurements. This loss function sacrifices the smoothness but enjoys advantages in statistical and computational efficiency. It also has potential to be extended in various scenarios. One interesting direction is to extend such an algorithm to exploit signal structures (e.g., nonnegativity, sparsity, etc) to assist the recovery. The lower-order loss function may offer great simplicity to prove performance guarantee in such cases. Another interesting topic is to study stochastic version of reshaped-WF. We have observed in preliminary experiments that the stochastic version of reshaped-WF converges fast numerically. It will be of great interest to fully understand the theoretic performance of such an algorithm and explore the reason behind its fast convergence.

**Acknowledgments**

This work is supported in part by the grants AFOSR FA9550-16-1-0077 and NSF ECCS 16-09916.

## Footnotes

[1]The loss function (3) was also used in Fienup [1982] to derive a gradient-like update for the phase retrieval problem with Fourier magnitude measurements. However, our paper is to characterize global convergence guarantee for such an algorithm with appropriate initialization, which was not studied in Fienup [1982].

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
