[Supplementary Material]

# Supplementary Material

We first introduce some notations here. We let $\mathcal{A} : \mathbb{R}^{n \times n} \mapsto \mathbb{R}^m$ be a linear map

$$\boldsymbol{M} \in \mathbb{R}^{n \times n} \mapsto \mathcal{A}(\boldsymbol{M}) := \{\boldsymbol{a}_i^T \boldsymbol{M} \boldsymbol{a}_i\}_{1 \leq i \leq m}.$$

We let $\| \cdot \|_1$ and $\| \cdot \|$ denote the $l_1$ norm and $l_2$ norm of a vector, respectively. Moreover, let $\| \cdot \|_F$ and $\| \cdot \|$ denote the Frobenius norm and the spectral norm of a matrix, respectively.

## A  Proof of Proposition 1: Performance Guarantee for Initialization

The arguments adapt the proof for truncated-WF Chen and Candes [2015] with further development to address measurements as absolute values and truncation from both sides.

We first estimate the norm of $\boldsymbol{x}$ as

$$\lambda_0 = \frac{mn}{\sum_{i=1}^m \|\boldsymbol{a}_i\|_1} \cdot \left( \frac{1}{m} \sum_{i=1}^m y_i \right). \tag{18}$$

Since $\boldsymbol{a}_i \sim \mathcal{N}(0, \boldsymbol{I}_{n \times n})$, by Hoeffding-type inequality, it can be shown that

$$\left| \frac{\sum_{i=1}^m \|\boldsymbol{a}_i\|_1}{mn} - \sqrt{\frac{2}{\pi}} \right| < \frac{\epsilon}{3} \tag{19}$$

holds with probability at least $1 - 2\exp(-c_1 mn\epsilon^2)$ for some constant $c_1 > 0$.

Moreover, given $\boldsymbol{x}$, $y_i$'s are independent sub-Gaussian random variables. Thus, by Hoeffding-type inequality, it can be shown that

$$\left| \sqrt{\frac{\pi}{2}} \left( \frac{1}{m} \sum_{i=1}^m y_i \right) - \|\boldsymbol{x}\| \right| < \frac{\epsilon}{3} \|\boldsymbol{x}\| \tag{20}$$

holds with probability at least $1 - 2\exp(-c_1 m\epsilon^2)$ for some constant $c_1 > 0$.

On the event $E_1 = \{\text{both (19) and (20) hold}\}$, it can be argued that

$$|\lambda_0 - \|\boldsymbol{x}\|| < \epsilon \|\boldsymbol{x}\|. \tag{21}$$

Without loss of generality, we let $\|\boldsymbol{x}\| = 1$. Then on the event $E_1$, the truncation function satisfies the following bounds

$$\mathbf{1}_{\{\alpha_l(1+\epsilon) < |\boldsymbol{a}_i^T \boldsymbol{x}| < \alpha_u(1-\epsilon)\}} \leq \mathbf{1}_{\{\alpha_l \lambda_0 < y_i < \alpha_u \lambda_0\}} \leq \mathbf{1}_{\{\alpha_l(1-\epsilon) < |\boldsymbol{a}_i^T \boldsymbol{x}| < \alpha_u(1+\epsilon)\}}.$$

Thus, by defining

$$\boldsymbol{Y}_1 := \frac{1}{m} \sum \boldsymbol{a}_i \boldsymbol{a}_i^T |\boldsymbol{a}_i^T \boldsymbol{x}| \mathbf{1}_{\{\alpha_l(1+\epsilon) < |\boldsymbol{a}_i^T \boldsymbol{x}| < \alpha_u(1-\epsilon)\}}$$

$$\boldsymbol{Y}_2 := \frac{1}{m} \sum \boldsymbol{a}_i \boldsymbol{a}_i^T |\boldsymbol{a}_i^T \boldsymbol{x}| \mathbf{1}_{\{\alpha_l(1-\epsilon) < |\boldsymbol{a}_i^T \boldsymbol{x}| < \alpha_u(1+\epsilon)\}},$$

we have $\boldsymbol{Y}_1 \prec \boldsymbol{Y} \prec \boldsymbol{Y}_2$. We further compute the expectations of $\boldsymbol{Y}_1$ and $\boldsymbol{Y}_2$ and obtain

$$\mathrm{E}[\boldsymbol{Y}_1] = (\beta_1 \boldsymbol{x}\boldsymbol{x}^T + \beta_2 \boldsymbol{I}), \quad \mathrm{E}[\boldsymbol{Y}_2] = (\beta_3 \boldsymbol{x}\boldsymbol{x}^T + \beta_4 \boldsymbol{I}), \tag{22}$$

where

$$\beta_1 := \mathrm{E}[|\xi|^3 \mathbf{1}_{\{\alpha_l(1+\epsilon) < |\xi| < \alpha_u(1-\epsilon)\}}] - \mathrm{E}[|\xi| \mathbf{1}_{\{\alpha_l(1+\epsilon) < |\xi| < \alpha_u(1-\epsilon)\}}],$$

$$\beta_2 := \mathrm{E}[|\xi| \mathbf{1}_{\{\alpha_l(1+\epsilon) < |\xi| < \alpha_u(1-\epsilon)\}}]$$

$$\beta_3 := \mathrm{E}[|\xi|^3 \mathbf{1}_{\{\alpha_l(1-\epsilon) < |\xi| < \alpha_u(1+\epsilon)\}}] - \mathrm{E}[|\xi| \mathbf{1}_{\{\alpha_l(1-\epsilon) < |\xi| < \alpha_u(1+\epsilon)\}}],$$

$$\beta_4 := \mathrm{E}[|\xi| \mathbf{1}_{\{\alpha_l(1-\epsilon) < |\xi| < \alpha_u(1+\epsilon)\}}]$$

where $\xi \sim \mathcal{N}(0, 1)$. For given $\alpha_l$ and $\alpha_u$, small value of $\epsilon$ yields arbitrarily close $\beta_1$ and $\beta_3$, as well as arbitrarily close $\beta_2$ and $\beta_4$. For example, taking $\alpha_l = 1, \alpha_u = 5$ and $\epsilon = 0.01$, we have $\beta_1 = 0.9678, \beta_2 = 0.4791, \beta_3 = 0.9678, \beta_4 = 0.4888$.

Now applying standard results on random matrices with non-isotropic sub-Gaussian rows Vershynin [2012, equation (5.26)] and noticing that $\boldsymbol{a}_i \boldsymbol{a}_i^T |\boldsymbol{a}_i^T \boldsymbol{x}| \mathbf{1}_{\{\alpha_l(1+\epsilon) < |\boldsymbol{a}_i^T \boldsymbol{x}| < \alpha_u(1-\epsilon)\}}$ can be rewritten as $\boldsymbol{b}_i \boldsymbol{b}_i^T$ for sub-Gaussian vector $\boldsymbol{b}_i := \boldsymbol{a}_i \sqrt{|\boldsymbol{a}_i^T \boldsymbol{x}|} \mathbf{1}_{\{\alpha_l(1+\epsilon) < |\boldsymbol{a}_i^T \boldsymbol{x}| < \alpha_u(1-\epsilon)\}}$, one can derive

$$\|\boldsymbol{Y}_1 - \mathrm{E}[\boldsymbol{Y}_1]\| \leq \delta, \quad \|\boldsymbol{Y}_2 - \mathrm{E}[\boldsymbol{Y}_2]\| \leq \delta \tag{23}$$

with probability $1 - 4\exp(-c_1(\delta)m)$ for some positive $c_1$ which is only affected by $\delta$, provided that $m/n$ exceeds a certain constant. Furthermore, when $\epsilon$ is sufficiently small, one further has $\|\mathrm{E}[\boldsymbol{Y}_1] - \mathrm{E}[\boldsymbol{Y}_2]\| \leq \delta$. Combining the above facts together, one can show that

$$\|\boldsymbol{Y} - (\beta_1 \boldsymbol{x}\boldsymbol{x}^T + \beta_2 \boldsymbol{I})\| \leq 3\delta. \tag{24}$$

Let $\tilde{\boldsymbol{z}}^{(0)}$ be the normalized leading eigenvector of $\boldsymbol{Y}$. Following the arguments in Candès et al. [2015, Section 7.8] and taking $\delta$ and $\epsilon$ to be sufficiently small, one has

$$\mathrm{dist}(\tilde{\boldsymbol{z}}^{(0)}, \boldsymbol{x}) \leq \tilde{\delta}, \tag{25}$$

for a given $\tilde{\delta} > 0$, as long as $m/n$ exceeds a certain constant.

# B  Supporting Arguments for Section 2.2

## B.1  Expectation of loss functions

The expectation of the loss function (2) of WF is given by Sun et al. [2016] as

$$\mathrm{E}[\ell_{WF}(\boldsymbol{z})] = \frac{3}{4}\|\boldsymbol{x}\|^4 + \frac{3}{4}\|\boldsymbol{z}\|^4 - \frac{1}{2}\|\boldsymbol{x}\|^2\|\boldsymbol{z}\|^2 - |\boldsymbol{z}^T\boldsymbol{x}|^2. \tag{26}$$

We next show that the expectation of the loss function (3) of reshaped-WF has the following form:

$$\mathrm{E}[\ell(\boldsymbol{z})] = \frac{1}{2}\|\boldsymbol{x}\|^2 + \frac{1}{2}\|\boldsymbol{z}\|^2 - \|\boldsymbol{x}\|\|\boldsymbol{z}\| \cdot \mathrm{E}\left[\frac{|\boldsymbol{a}_i^T\boldsymbol{z}|}{\|\boldsymbol{z}\|} \cdot \frac{|\boldsymbol{a}_i^T\boldsymbol{x}|}{\|\boldsymbol{x}\|}\right], \tag{27}$$

where

$$\mathrm{E}\left[\frac{|\boldsymbol{a}_i^T\boldsymbol{z}|}{\|\boldsymbol{z}\|} \cdot \frac{|\boldsymbol{a}_i^T\boldsymbol{x}|}{\|\boldsymbol{x}\|}\right] = \begin{cases} \frac{(1-\rho^2)^{3/2}}{\pi}\int_0^\infty t(e^{\rho t} + e^{-\rho t})K_0(t)dt, & \text{if } |\rho| < 1; \\ 1, & \text{if } |\rho| = 1; \end{cases} \tag{28}$$

where $\rho = \frac{\boldsymbol{z}^T\boldsymbol{x}}{\|\boldsymbol{x}\|\|\boldsymbol{z}\|}$ and $K_0(\cdot)$ is the modified Bessel function of the second kind.

In order to derive (28), we first define

$$u := \frac{\boldsymbol{a}_i^T\boldsymbol{z}}{\|\boldsymbol{z}\|} \text{ and } v := \frac{\boldsymbol{a}_i^T\boldsymbol{x}}{\|\boldsymbol{x}\|},$$

and it suffices to drive $\mathrm{E}[|uv|]$. Note that $(u, v) \sim \mathcal{N}(0, \Sigma)$, where

$$\Sigma = \begin{bmatrix} 1 & \rho \\ \rho & 1 \end{bmatrix}, \quad \text{and} \quad \rho = \frac{\boldsymbol{z}^T\boldsymbol{x}}{\|\boldsymbol{x}\|\|\boldsymbol{z}\|}.$$

Following Donahue [1964], the density function of $u \cdot v$ is given by

$$\phi_{uv}(x) = \frac{1}{\pi\sqrt{1-\rho^2}}\exp\left(\frac{\rho x}{1-\rho^2}\right)K_0\left(\frac{|x|}{1-\rho^2}\right), \quad x \neq 0.$$

Thus, the density of $|uv|$ is given by

$$\psi_{|uv|}(x) = \frac{1}{\pi\sqrt{1-\rho^2}}\left[\exp\left(\frac{\rho x}{1-\rho^2}\right) + \exp\left(-\frac{\rho x}{1-\rho^2}\right)\right]K_0\left(\frac{|x|}{1-\rho^2}\right), \quad x > 0, \tag{29}$$

for $|\rho| < 1$. Therefore, if $|\rho| < 1$, then

$$\begin{aligned}
\mathrm{E}[|uv|] &= \int_0^\infty x \cdot \psi_\rho(x)dx \\
&= \int_0^\infty x \cdot \frac{1}{\pi\sqrt{1-\rho^2}}\left[\exp\left(\frac{\rho x}{1-\rho^2}\right) + \exp\left(-\frac{\rho x}{1-\rho^2}\right)\right]K_0\left(\frac{|x|}{1-\rho^2}\right)dx \\
&= \frac{(1-\rho^2)^{3/2}}{\pi}\int_0^\infty t(e^{\rho t} + e^{-\rho t})K_0(t)dt
\end{aligned}$$

where the last step follows by changing variables.

If $|\rho| = 1$, then $|uv|$ becomes a $\chi_1^2$ random variable, with the density

$$\psi_{|uv|}(x) = \frac{1}{\sqrt{2\pi}}x^{-1/2}\exp(-x/2), \quad x > 0,$$

and hence $\mathrm{E}[|uv|] = 1$.

## B.2 Proof of Lemma 1

Let $\boldsymbol{a}(1)$ denote the first element of a generic vector $\boldsymbol{a}$, and $\boldsymbol{a}(-1)$ denote the remaining vector of $\boldsymbol{a}$ after eliminating the first element. Let $\boldsymbol{U}_x$ be an orthonormal matrix with first row being $\boldsymbol{x}^T/\|\boldsymbol{x}\|$, $\tilde{\boldsymbol{a}}_i = \boldsymbol{U}_x\boldsymbol{a}_i$, and $\tilde{\boldsymbol{h}} = \boldsymbol{U}_x\boldsymbol{h}$. Similarly define $\boldsymbol{U}_{\tilde{h}(-1)}$ and let $\tilde{\boldsymbol{b}}_i = \boldsymbol{U}_{\tilde{h}(-1)}\tilde{\boldsymbol{a}}_i(-1)$. Then $\tilde{\boldsymbol{a}}_i(1)$ and $\tilde{\boldsymbol{b}}_i(1)$ are independent standard Gaussian random variables.

We evaluate the conditional probability as follows.

$$
\begin{aligned}
&\mathbb{P}\{(\boldsymbol{a}_i^T\boldsymbol{x})(\boldsymbol{a}_i^T\boldsymbol{z}) < 0 \big| (\boldsymbol{a}_i^T\boldsymbol{x})^2 = t\|\boldsymbol{x}\|^2\} \\
&= \mathbb{P}\{t\|\boldsymbol{x}\|^2 + (\boldsymbol{a}_i^T\boldsymbol{x})(\boldsymbol{a}_i^T\boldsymbol{h}) < 0 \big| (\boldsymbol{a}_i^T\boldsymbol{x})^2 = t\|\boldsymbol{x}\|^2\} \quad \text{due to } \boldsymbol{z} = \boldsymbol{x} + \boldsymbol{h} \\
&\leq \mathbb{P}\{t\|\boldsymbol{x}\|^2 - \sqrt{t}\|\boldsymbol{x}\||\boldsymbol{a}_i^T\boldsymbol{h}| < 0 \big| (\boldsymbol{a}_i^T\boldsymbol{x})^2 = t\|\boldsymbol{x}\|^2\} \\
&= \mathbb{P}\{|\boldsymbol{a}_i^T\boldsymbol{h}| > \sqrt{t}\|\boldsymbol{x}\| \big| (\boldsymbol{a}_i^T\boldsymbol{x})^2 = t\|\boldsymbol{x}\|^2\} \\
&= \mathbb{P}\{|\tilde{\boldsymbol{a}}_i(1)\tilde{\boldsymbol{h}}(1) + \tilde{\boldsymbol{a}}_i(-1)^T\tilde{\boldsymbol{h}}(-1)| > \sqrt{t}\|\boldsymbol{x}\| \big| |\tilde{\boldsymbol{a}}_i(1)| = \sqrt{t}\} \quad \text{orthogonal transformation } U_x \\
&\leq \mathbb{P}\{|\tilde{\boldsymbol{a}}_i(1)\tilde{\boldsymbol{h}}(1)| + |\tilde{\boldsymbol{a}}_i(-1)^T\tilde{\boldsymbol{h}}(-1)| > \sqrt{t}\|\boldsymbol{x}\| \big| |\tilde{\boldsymbol{a}}_i(1)| = \sqrt{t}\} \\
&= \mathbb{P}\left\{|\tilde{\boldsymbol{a}}_i(-1)^T\tilde{\boldsymbol{h}}(-1)| > \sqrt{t}\left(\|\boldsymbol{x}\| - \frac{|\boldsymbol{h}^T\boldsymbol{x}|}{\|\boldsymbol{x}\|}\right) \Big| |\tilde{\boldsymbol{a}}_i(1)| = \sqrt{t}\right\} \quad \text{due to } \tilde{\boldsymbol{h}}(1) = \frac{\boldsymbol{h}^T\boldsymbol{x}}{\|\boldsymbol{x}\|} \\
&= \mathbb{P}\left\{|\boldsymbol{b}_i(1)| \cdot \sqrt{\|\boldsymbol{h}\|^2 - \frac{(\boldsymbol{h}^T\boldsymbol{x})^2}{\|\boldsymbol{x}\|^2}} > \sqrt{t}\left(\|\boldsymbol{x}\| - \frac{|\boldsymbol{h}^T\boldsymbol{x}|}{\|\boldsymbol{x}\|}\right)\right\} \quad \text{due to } \boldsymbol{b} = U_{\tilde{h}(-1)}\tilde{\boldsymbol{a}}_i(-1) \\
&= \mathbb{P}\left\{|\boldsymbol{b}_i(1)| > \sqrt{t} \cdot \frac{\|\boldsymbol{x}\|}{\|\boldsymbol{h}\|}\left(1 - \frac{|\boldsymbol{h}^T\boldsymbol{x}|}{\|\boldsymbol{x}\|^2}\right) \Big/ \sqrt{1 - (\boldsymbol{h}^T\boldsymbol{x})^2/(\|\boldsymbol{h}\|^2\|\boldsymbol{x}\|^2)}\right\} \\
&\leq \mathbb{P}\left\{|\boldsymbol{b}_i(1)| > \sqrt{t} \cdot \frac{\|\boldsymbol{x}\|}{\|\boldsymbol{h}\|}\left(1 - \frac{|\boldsymbol{h}^T\boldsymbol{x}|}{\|\boldsymbol{x}\|^2}\right)\right\} \\
&\leq \mathbb{P}\left\{|\boldsymbol{b}_i(1)| > \sqrt{t} \cdot \left(\frac{\|\boldsymbol{x}\|}{\|\boldsymbol{h}\|} - 1\right)\right\} \quad \text{by Cauchy-Schwartz inequality} \\
&\leq \mathbb{P}\left\{|\boldsymbol{b}_i(1)/\sqrt{2}| > \frac{\sqrt{t}}{\sqrt{2}} \cdot \left(\frac{\|\boldsymbol{x}\|}{\|\boldsymbol{h}\|} - 1\right)\right\} \\
&\leq \operatorname{erfc}\left(\frac{\sqrt{t}\|\boldsymbol{x}\|}{2\|\boldsymbol{h}\|}\right)
\end{aligned}
$$

where $\operatorname{erfc}(z) := \frac{2}{\sqrt{\pi}}\int_z^\infty \exp(-t^2)dt$, and the last inequality holds if $\|\boldsymbol{h}\| < (1 - 1/\sqrt{2})\|\boldsymbol{x}\|$.

# C  Proof of Theorem 1: Geometric Convergence of Reshaped-WF

The general structure of the proof follows that for WF in Candès et al. [2015] and truncated-WF in Chen and Candes [2015]. However, the proof is much simpler due to the lower-order loss function adopted in reshaped-WF. The proof also requires development of new bounds due to the nonsmoothness of the loss function and absolute value based measurements.

The idea of the proof is to show that within the neighborhood of global optimums, reshaped-WF satisfies the *Regularity Condition* $\mathsf{RC}(\mu, \lambda, c)$, i.e.,

$$\langle\nabla\ell(\boldsymbol{z}), \boldsymbol{h}\rangle \geq \frac{\mu}{2}\|\nabla\ell(\boldsymbol{z})\|^2 + \frac{\lambda}{2}\|\boldsymbol{h}\|^2 \tag{30}$$

for all $\boldsymbol{z}$ and $\boldsymbol{h} = \boldsymbol{z} - \boldsymbol{x}$ obeying $\|\boldsymbol{h}\| \leq c\|\boldsymbol{x}\|$, where $0 < c < 1$ is some constant. Then, as shown in Chen and Candes [2015], once the initialization lands into this neighborhood, geometric convergence can be guaranteed, i.e.,

$$\operatorname{dist}^2(\boldsymbol{z} + \mu\nabla\ell(\boldsymbol{z}), \boldsymbol{x}) \leq (1 - \mu\lambda)\operatorname{dist}^2(\boldsymbol{z}, \boldsymbol{x}), \tag{31}$$

for any $\boldsymbol{z}$ with $\|\boldsymbol{z} - \boldsymbol{x}\| \leq c\|\boldsymbol{x}\|$.

To show the regularity condition, we first define a set $\mathcal{S} := \{i : 1 \leq i \leq m, (\boldsymbol{a}_i^T\boldsymbol{z})(\boldsymbol{a}_i^T\boldsymbol{x}) < 0\}$, and then derive the following bound:

$$
\begin{aligned}
\langle\nabla\ell(\boldsymbol{z}), \boldsymbol{h}\rangle &= \frac{1}{m}\sum_{i=1}^m \left(\boldsymbol{a}_i^T\boldsymbol{z} - |\boldsymbol{a}_i^T\boldsymbol{x}|\operatorname{sgn}(\boldsymbol{a}_i^T\boldsymbol{z})\right)(\boldsymbol{a}_i^T\boldsymbol{h}) = \frac{1}{m}\left[\sum_{i=1}^m(\boldsymbol{a}_i^T\boldsymbol{h})^2 + 2\sum_{i\in\mathcal{S}}(\boldsymbol{a}_i^T\boldsymbol{x})(\boldsymbol{a}_i^T\boldsymbol{h})\right] \\
&\geq \frac{1}{m}\left[\sum_{i=1}^m(\boldsymbol{a}_i^T\boldsymbol{h})^2 - 2\left|\sum_{i\in\mathcal{S}}(\boldsymbol{a}_i^T\boldsymbol{x})(\boldsymbol{a}_i^T\boldsymbol{h})\right|\right] \geq \frac{1}{m}\left[\sum_{i=1}^m(\boldsymbol{a}_i^T\boldsymbol{h})^2 - \sum_{i\in\mathcal{S}}2\left|(\boldsymbol{a}_i^T\boldsymbol{x})(\boldsymbol{a}_i^T\boldsymbol{h})\right|\right]. \tag{32}
\end{aligned}
$$

The first term in (32) can be bounded using Lemma 3.1 in Candès et al. [2013], which we state below.

**Lemma 2.** *For any $0 < \epsilon < 1$, if $m > c_0 n \epsilon^{-2} \log \epsilon^{-1}$, then with probability at least $1 - \exp(-\epsilon^2 m/8)$,*

$$\frac{1}{m} \sum_{i=1}^{m} (\boldsymbol{a}_i^T \boldsymbol{h})^2 \geq (1 - \epsilon) \|\boldsymbol{h}\|^2 \tag{33}$$

*holds for all non-zero vectors $\boldsymbol{h} \in \mathbb{R}^n$.*

For the second term in (32), we derive

$$
\begin{aligned}
\sum_{i \in \mathcal{S}} 2 \left| \boldsymbol{a}_i^T \boldsymbol{x} \right| \left| \boldsymbol{a}_i^T \boldsymbol{h} \right| &\leq \sum_{i \in \mathcal{S}} \left[ (\boldsymbol{a}_i^T \boldsymbol{x})^2 + (\boldsymbol{a}_i^T \boldsymbol{h})^2 \right] \\
&= \sum_{i=1}^{m} [(\boldsymbol{a}_i^T \boldsymbol{x})^2 + (\boldsymbol{a}_i^T \boldsymbol{h})^2] \cdot \mathbf{1}_{\{(\boldsymbol{a}_i^T \boldsymbol{x})(\boldsymbol{a}_i^T \boldsymbol{z}) < 0\}} \\
&= \sum_{i=1}^{m} [(\boldsymbol{a}_i^T \boldsymbol{x})^2 + (\boldsymbol{a}_i^T \boldsymbol{h})^2] \cdot \mathbf{1}_{\{(\boldsymbol{a}_i^T \boldsymbol{x})^2 + (\boldsymbol{a}_i^T \boldsymbol{x})(\boldsymbol{a}_i^T \boldsymbol{h}) < 0\}} \\
&\leq \sum_{i=1}^{m} [(\boldsymbol{a}_i^T \boldsymbol{x})^2 + (\boldsymbol{a}_i^T \boldsymbol{h})^2] \cdot \mathbf{1}_{\{|\boldsymbol{a}_i^T \boldsymbol{x}| < |\boldsymbol{a}_i^T \boldsymbol{h}|\}} \\
&\leq 2 \sum_{i=1}^{m} (\boldsymbol{a}_i^T \boldsymbol{h})^2 \cdot \mathbf{1}_{\{|\boldsymbol{a}_i^T \boldsymbol{x}| < |\boldsymbol{a}_i^T \boldsymbol{h}|\}}.
\end{aligned}
\tag{34}
$$

The above equation can be further upper bounded by the following lemma.

**Lemma 3.** *For any $\epsilon > 0$, if $m > c_0 n \epsilon^{-2} \log \epsilon^{-1}$, then with probability at least $1 - C \exp(-c_1 \epsilon^2 m)$,*

$$\frac{1}{m} \sum_{i=1}^{m} (\boldsymbol{a}_i^T \boldsymbol{h})^2 \cdot \mathbf{1}_{\{|\boldsymbol{a}_i^T \boldsymbol{x}| < |\boldsymbol{a}_i^T \boldsymbol{h}|\}} \leq (0.13 + \epsilon) \|\boldsymbol{h}\|^2 \tag{35}$$

*holds for all non-zero vectors $\boldsymbol{h} \in \mathbb{R}^n$ satisfying $\|\boldsymbol{h}\| \leq \frac{1}{10} \|\boldsymbol{x}\|$. Here, $c_0, c_1, C > 0$ are some universal constants.*

*Proof.* See Section C.1. □

Therefore, combining Lemmas 2 and 3 with (32) yields

$$\langle \nabla \ell(\boldsymbol{z}), \boldsymbol{h} \rangle \geq (1 - 0.26 - 3\epsilon) \|\boldsymbol{h}\|^2 = (0.74 - 3\epsilon) \|\boldsymbol{h}\|^2. \tag{36}$$

We further provide an upper bound on $\|\nabla \ell(\boldsymbol{z})\|$ in the following lemma.

**Lemma 4.** *Fix $\delta > 0$, and assume $y_i = |\boldsymbol{a}_i^T \boldsymbol{x}|$. Suppose that $m \geq c_0 n$ for a certain constant $c_0 > 0$. There exist some universal constants $c, C > 0$ such that with probability at least $1 - C \exp(-cm)$,*

$$\|\nabla \ell(\boldsymbol{z})\| \leq (1 + \delta) \cdot 2 \|\boldsymbol{h}\| \tag{37}$$

*holds for all non-zero vectors $\boldsymbol{h}, \boldsymbol{z} \in \mathbb{R}^n$ satisfying $\boldsymbol{z} = \boldsymbol{x} + \boldsymbol{h}$ and $\frac{\|\boldsymbol{h}\|}{\|\boldsymbol{x}\|} \leq \frac{1}{10}$.*

*Proof.* See Section C.2. □

Thus, applying Lemma 4 to (36), we conclude that *Regularity Condition* (30) holds for $\mu$ and $\lambda$ satisfying

$$0.74 - 2\epsilon \geq \frac{\mu}{2} \cdot 4(1 + \delta)^2 + \frac{\lambda}{2}, \tag{38}$$

which concludes the proof. The proofs of two major lemmas are provided in the following two subsections.

## C.1 Proof of Lemma 3

We first prove bounds for any fixed $\boldsymbol{h} \leq \frac{1}{10} \|\boldsymbol{x}\|$, and then develop a uniform bound later on. We introduce a series of auxiliary random Lipschitz functions to approximate the indicator functions. For $i = 1, \ldots, m$, define

$$\chi_i(t) := \begin{cases} t, & \text{if } t > (\boldsymbol{a}_i^T \boldsymbol{x})^2; \\ \frac{1}{\delta}(t - (\boldsymbol{a}_i^T \boldsymbol{x})^2) + (\boldsymbol{a}_i^T \boldsymbol{x})^2, & \text{if } (1 - \delta)(\boldsymbol{a}_i^T \boldsymbol{x})^2 \leq t \leq (\boldsymbol{a}_i^T \boldsymbol{x})^2; \\ 0, & \text{else}; \end{cases} \tag{39}$$

and then $\chi_i(t)$'s are random Lipschitz functions with Lipschitz constant $\frac{1}{\delta}$. We further have

$$|\boldsymbol{a}_i^T \boldsymbol{h}|^2 \mathbf{1}_{\{|\boldsymbol{a}_i^T \boldsymbol{x}| < |\boldsymbol{a}_i^T \boldsymbol{h}|\}} \leq \chi_i(|\boldsymbol{a}_i^T \boldsymbol{h}|^2) \leq |\boldsymbol{a}_i^T \boldsymbol{h}|^2 \mathbf{1}_{\{(1-\delta)|\boldsymbol{a}_i^T \boldsymbol{x}|^2 < |\boldsymbol{a}_i^T \boldsymbol{h}|^2\}}. \tag{40}$$

For convenience, we denote $\gamma_i := \frac{|\boldsymbol{a}_i^T \boldsymbol{h}|^2}{\|\boldsymbol{h}\|^2} \mathbf{1}_{\{(1-\delta)|\boldsymbol{a}_i^T \boldsymbol{x}|^2 < |\boldsymbol{a}_i^T \boldsymbol{h}|^2\}}$ and $\theta := \|\boldsymbol{h}\|/\|\boldsymbol{x}\|$. We next estimate the expectation of $\gamma_i$, by conditional expectation,

$$\mathrm{E}[\gamma_i] = \int_{\Omega} \gamma_i d\mathbb{P} = \iint_{-\infty}^{\infty} \mathrm{E}\left[\gamma_i \big| \boldsymbol{a}_i^T \boldsymbol{x} = \tau_1 \|\boldsymbol{x}\|, \boldsymbol{a}_i^T \boldsymbol{h} = \tau_2 \|\boldsymbol{h}\|\right] \cdot f(\tau_1, \tau_2) d\tau_1 d\tau_2, \tag{41}$$

where $f(\tau_1, \tau_2)$ is the density of two joint Gaussian random variables with correlation $\rho = \frac{\boldsymbol{h}^T \boldsymbol{x}}{\|\boldsymbol{h}\|\|\boldsymbol{x}\|} \neq \pm 1$. We then continue to derive

$$\mathrm{E}[\gamma_i] = \iint_{-\infty}^{\infty} \tau_2^2 \cdot \mathbf{1}_{\{\sqrt{1-\delta}|\tau_1| < |\tau_2|\theta\}} \cdot f(\tau_1, \tau_2) d\tau_1 d\tau_2$$

$$= \frac{1}{2\pi\sqrt{1-\rho^2}} \int_{-\infty}^{\infty} \tau_2^2 \exp\left(-\frac{\tau_2^2}{2}\right) \cdot \int_{\frac{-|\tau_2|\theta}{\sqrt{1-\delta}}}^{\frac{|\tau_2|\theta}{\sqrt{1-\delta}}} \exp\left(-\frac{(\tau_1 - \rho\tau_2)^2}{2(1-\rho^2)}\right) d\tau_1 d\tau_2 \tag{42}$$

$$= \frac{1}{2\pi} \int_{-\infty}^{\infty} \tau_2^2 \exp\left(-\frac{\tau_2^2}{2}\right) \cdot \int_{-\frac{\frac{|\tau_2|\theta}{\sqrt{1-\delta}} - \rho\tau_2}{\sqrt{1-\rho^2}}}^{\frac{\frac{|\tau_2|\theta}{\sqrt{1-\delta}} - \rho\tau_2}{\sqrt{1-\rho^2}}} \exp\left(-\frac{\tau^2}{2}\right) d\tau d\tau_2 \qquad \text{by changing variables}$$

$$= \frac{1}{2\pi} \int_{-\infty}^{\infty} \tau_2^2 \exp\left(-\frac{\tau_2^2}{2}\right) \cdot \sqrt{\frac{\pi}{2}} \left(\mathrm{erf}\left(\frac{\frac{|\tau_2|\theta}{\sqrt{1-\delta}} - \rho\tau_2}{\sqrt{2(1-\rho^2)}}\right) - \mathrm{erf}\left(\frac{-\frac{|\tau_2|\theta}{\sqrt{1-\delta}} - \rho\tau_2}{\sqrt{2(1-\rho^2)}}\right)\right) d\tau_2$$

$$= \frac{1}{\sqrt{2\pi}} \int_0^{\infty} \tau_2^2 \exp\left(-\frac{\tau_2^2}{2}\right) \cdot \left(\mathrm{erf}\left(\frac{(\frac{\theta}{\sqrt{1-\delta}} - \rho)\tau_2}{\sqrt{2(1-\rho^2)}}\right) + \mathrm{erf}\left(\frac{(\frac{\theta}{\sqrt{1-\delta}} + \rho)\tau_2}{\sqrt{2(1-\rho^2)}}\right)\right) d\tau_2. \tag{43}$$

For $|\rho| < 1$, $\mathrm{E}[\gamma_i]$ is a continuous function of $\rho$. For $|\rho| = 1$, $\mathrm{E}[\gamma_i] = 0$. The last integral (43) can be calculated numerically. Figure 5 plots $\mathrm{E}[\gamma_i]$ for $\theta = 0.1$ and $\delta = 0.01$ over $\rho \in [-1, 1]$. Furthermore, (42) indicates that $\mathrm{E}[\gamma_i]$ is monotonically increasing with both $\theta$ and $\delta$. Thus, we obtain a universal bound

$$\mathrm{E}[\gamma_i] \leq 0.13 \quad \text{for } \theta < 0.1 \text{ and } \delta = 0.01, \tag{44}$$

which further implies $\mathrm{E}[\chi_i(|\boldsymbol{a}_i^T \boldsymbol{h}|^2)] \leq 0.13\|\boldsymbol{h}\|^2$ for $\theta < 0.1$ and $\delta = 0.01$.

Figure 5: $\mathrm{E}[\gamma_i]$ with respect to $\rho$

Furthermore, $\chi_i(|\boldsymbol{a}_i^T \boldsymbol{h}|^2)$'s are sub-exponential with sub-exponential norm $\mathcal{O}(\|\boldsymbol{h}\|^2)$. By the sub-exponential tail bound (Bernstein type) Vershynin [2012], we have

$$\mathcal{P}\left[\frac{1}{m} \sum_{i=1}^m \frac{\chi_i(|\boldsymbol{a}_i^T \boldsymbol{h}|^2)}{\|\boldsymbol{h}\|^2} > (0.13 + \epsilon)\right] < \exp(-cm\epsilon^2), \tag{45}$$

for some universal constant $c$, as long as $\|\boldsymbol{h}\| \leq \frac{1}{10}\|\boldsymbol{x}\|$.

We have proved so far that the claim holds for a fixed $\boldsymbol{h}$. We next obtain a uniform bound over all $\boldsymbol{h}$ satisfying $\|\boldsymbol{h}\| \leq \frac{1}{10}\|\boldsymbol{x}\|$. We first show the claim holds when $\|\boldsymbol{h}\| = \frac{1}{10}\|\boldsymbol{x}\|$ and then argue the claim holds when $\|\boldsymbol{h}\| < \frac{1}{10}\|\boldsymbol{x}\|$ towards the end of the proof. Let $\epsilon' = \epsilon\frac{\|\boldsymbol{x}\|}{10}$ and we construct an $\epsilon'-$net $\mathcal{N}_{\epsilon'}$ covering the sphere with radius $\frac{1}{10}\|\boldsymbol{x}\|$ in $\mathbb{R}^n$ with cardinality $|\mathcal{N}_{\epsilon'}| \leq (1 + \frac{2}{\epsilon})^n$. Then for any $\|\boldsymbol{h}\| = \frac{1}{10}\|\boldsymbol{x}\|$, there exists a $\boldsymbol{h}_0 \in \mathcal{N}_{\epsilon'}$ such that $\|\boldsymbol{h} - \boldsymbol{h}_0\| \leq \epsilon\|\boldsymbol{h}\|$. Taking the union bound for all the points on the net, we claim that

$$\frac{1}{m}\sum_{i=1}^m \chi_i\left(|\boldsymbol{a}_i^T\boldsymbol{h}_0|^2\right) \leq (0.13 + \epsilon)\|\boldsymbol{h}_0\|^2, \quad \forall \boldsymbol{h}_0 \in \mathcal{N}_{\epsilon'} \tag{46}$$

holds with probability at least $1 - (1 + 2/\epsilon)^n \exp(-cm\epsilon^2)$.

Since $\chi_i(t)$'s are Lipschitz functions with constant $1/\delta$, we have the following bound

$$\left|\chi_i(|\boldsymbol{a}_i^T\boldsymbol{h}|^2) - \chi_i(|\boldsymbol{a}_i^T\boldsymbol{h}_0|^2)\right| \leq \frac{1}{\delta}\left||\boldsymbol{a}_i^T\boldsymbol{h}|^2 - |\boldsymbol{a}_i^T\boldsymbol{h}_0|^2\right|. \tag{47}$$

Moreover, by Chen and Candes [2015, Lemma 1], we have

$$\frac{1}{m}\|\mathcal{A}(\boldsymbol{M})\|_1 \leq c_2\|\boldsymbol{M}\|_F, \qquad \text{for all symmetric rank-2 matrices } \boldsymbol{M} \in \mathbb{R}^{n \times n}, \tag{48}$$

holds with probability at least $1 - C\exp(-c_1 m)$ as long as $m > c_0 n$ for some constants $C, c_0, c_1, c_2 > 0$. Consequently, on the event that (48) holds, we have

$$\left|\frac{1}{m}\sum_{i=1}^m \chi_i\left(|\boldsymbol{a}_i^T\boldsymbol{h}|^2\right) - \frac{1}{m}\sum_{i=1}^m \chi_i\left(|\boldsymbol{a}_i^T\boldsymbol{h}_0|^2\right)\right|$$

$$\leq \frac{1}{m}\sum_{i=1}^m \left|\chi_i\left(|\boldsymbol{a}_i^T\boldsymbol{h}|^2\right) - \chi_i\left(|\boldsymbol{a}_i^T\boldsymbol{h}_0|^2\right)\right|$$

$$\leq \frac{1}{\delta} \cdot \frac{1}{m}\|\mathcal{A}(\boldsymbol{h}\boldsymbol{h}^T - \boldsymbol{h}_0\boldsymbol{h}_0^T)\|_1 \qquad \text{because of (47)}$$

$$\leq \frac{1}{\delta} \cdot c_2\|\boldsymbol{h}\boldsymbol{h}^T - \boldsymbol{h}_0\boldsymbol{h}_0^T\|_F \qquad \text{because of (48)}$$

$$\leq \frac{1}{\delta} \cdot 3c_2\|\boldsymbol{h} - \boldsymbol{h}_0\| \cdot \|\boldsymbol{h}\| \leq 3c_3\epsilon/\delta\|\boldsymbol{h}\|^2,$$

where the last inequality is due to the Lemma 2 in Chen and Candes [2015].

On the intersection of events that (46) and (48) hold, we have

$$\frac{1}{m}\sum_{i=1}^m \chi_i\left(|\boldsymbol{a}_i^T\boldsymbol{h}|^2\right) \leq (0.13 + \epsilon + 3c_3\epsilon/\delta)\|\boldsymbol{h}\|^2, \tag{49}$$

for all $\boldsymbol{h}$ with $\|\boldsymbol{h}\| = \frac{1}{10}\|\boldsymbol{x}\|$.

For the case when $\|\boldsymbol{h}'\| < \frac{1}{10}\|\boldsymbol{x}\|$, $\boldsymbol{h}' = \omega\boldsymbol{h}$ for some $\boldsymbol{h}$ satisfying $\|\boldsymbol{h}\| = \frac{1}{10}\|\boldsymbol{x}\|$ and $0 < \omega < 1$. By the definition of $\chi_i(\cdot)$, it can be verified that

$$\chi_i(|\boldsymbol{a}_i^T\boldsymbol{h}'|^2) = \chi_i(|\boldsymbol{a}_i^T(\omega\boldsymbol{h})|^2) \leq \omega^2\chi_i(|\boldsymbol{a}_i^T\boldsymbol{h}|^2). \tag{50}$$

Applying (49), on the same event that (46) and (48) hold, we have

$$\frac{1}{m}\sum_{i=1}^m \chi_i\left(|\boldsymbol{a}_i^T\boldsymbol{h}'|^2\right) \leq (0.13 + \epsilon + 3c_3\epsilon/\delta)\|\boldsymbol{h}'\|^2, \tag{51}$$

for all $\|\boldsymbol{h}'\| < \frac{1}{10}\|\boldsymbol{x}\|$. Since $\epsilon$ can be arbitrarily small, the proof is completed.

## C.2 Proof of Lemma 4

Denote $v_i := \boldsymbol{a}_i^T\boldsymbol{z} - |\boldsymbol{a}_i^T\boldsymbol{x}|\text{sgn}(\boldsymbol{a}_i^T\boldsymbol{z})$. Then

$$\nabla\ell(\boldsymbol{z}) = \frac{1}{m}\boldsymbol{A}^T\boldsymbol{v}, \tag{52}$$

where $\boldsymbol{A}$ is a matrix with each row being $\boldsymbol{a}_i^T$ and $\boldsymbol{v}$ is a $m-$dimensional vector with each entry being $v_i$. Thus,

$$\|\nabla\ell(\boldsymbol{z})\| = \left\|\frac{1}{m}\boldsymbol{A}^T\boldsymbol{v}\right\| \leq \frac{1}{m}\|\boldsymbol{A}\| \cdot \|\boldsymbol{v}\| \leq (1 + \delta)\frac{\|\boldsymbol{v}\|}{\sqrt{m}} \tag{53}$$

as long as $m \geq c_1 n$ for some sufficiently large $c_1 > 0$, where the spectral norm bound $\|\boldsymbol{A}\| \leq \sqrt{m}(1 + \delta)$ follows from Vershynin [2012, Theorem 5.32].

We next bound $\|\boldsymbol{v}\|$. Let $\boldsymbol{v} = \boldsymbol{v}^{(1)} + \boldsymbol{v}^{(2)}$, where $v_i^{(1)} = \boldsymbol{a}_i^T \boldsymbol{h}$ and $v_i^{(2)} = 2\boldsymbol{a}_i^T \boldsymbol{x} \mathbf{1}_{\{(\boldsymbol{a}_i^T \boldsymbol{z})(\boldsymbol{a}_i^T \boldsymbol{x})<0\}}$. By triangle inequality, we have $\|\boldsymbol{v}\| \leq \|\boldsymbol{v}^{(1)}\| + \|\boldsymbol{v}^{(2)}\|$. Furthermore, given $m > c_0 n$, by Candès et al. [2013, Lemma 3.1] with probability $1 - \exp(-cm)$, we have

$$\frac{1}{m}\|\boldsymbol{v}^{(1)}\|^2 = \frac{1}{m}\sum_{i=1}^{m}(\boldsymbol{a}_i^T \boldsymbol{h})^2 \leq (1+\delta)\|\boldsymbol{h}\|^2. \tag{54}$$

By Lemma 3, we have with probability $1 - C\exp(-c_1 m)$

$$\frac{1}{m}\|\boldsymbol{v}^{(2)}\|^2 = \frac{1}{m}\sum_{i=1}^{m} 4(\boldsymbol{a}_i^T \boldsymbol{x})^2 \cdot \mathbf{1}_{\{(\boldsymbol{a}_i^T \boldsymbol{x})(\boldsymbol{a}_i^T \boldsymbol{z})<0\}} \leq 4(0.13 + \epsilon)\|\boldsymbol{h}\|^2. \tag{55}$$

Hence,

$$\frac{\|\boldsymbol{v}\|}{\sqrt{m}} \leq [\sqrt{1+\delta} + 2\sqrt{0.13 + \epsilon}]\|\boldsymbol{h}\|. \tag{56}$$

This concludes the proof.

# D    Image of Milky Galaxy

Figure 6: Milky way Galaxy.