[Reviews · NeurIPS 2016]

Reviewer 1

Summary

This paper presents a new algorithm for the phase retrieval problem, which improves the truncated Wirtinger flow (TWF) algorithm of Chen and Candes. The proposed algorithm replaces a quadratic with an absolute value, and a gradient with a subgradient; it is proved to have the same statistical efficiency and computational speed as TWF, while being somewhat simpler, and performing a bit better in some of the numerical experiments.

Qualitative Assessment

EDIT: In light of the author feedback, I am increasing my score on the quality and novelty dimensions. The authors have convinced me that the analysis of the non-smooth loss is novel, that the performance in practice is substantially better than TWF on large problems, and that their approach requires tweaking fewer parameters. -- The observation that it's not necessary to square the magnitude of the observation to optimize the loss function, and that it may be better not to square, is nice: this is in keeping with much of the optimization literature over the last 30 years, which has learned that it's often better *not* to square nonsmooth objectives. The analysis the authors present to help readers informally understand what their gradient descent method does is very helpful and well written: figure 2, in particular, is really cool. However, the improvements in this paper, compared with TWF, seem minimal. The improvement in numerical performance doesn't seem overwhelming, and the proof strategy, as far as I can tell, is similar. Comments: * What you call \nabla \ell(z) should be properly called a(n element drawn from the) *subgradient*. * You still truncate to initialize your algorithm. Why is this necessary? * Line 212: actually, setting the step size can be a real problem in practice: I've observed WF to diverge sometimes! * Line 226: including a proof of robustness would be nice if you want to make this claim. * Figure 3a: looks like reshaped WF is not better? * a few grammar mistakes: eg, "much less number of iterations" should be "many fewer iterations"

Confidence in this Review

2-Confident (read it all; understood it all reasonably well)


Reviewer 2

Summary

This paper investigates the power of the Wirtinger-flow-type algorithm with a new objective function. By choosing a non-smooth yet lower-order objective function, the authors are able to achieve optimal computational complexity and statistical accuracy without enforcing truncation procedures as adopted in prior works. The authors also develop a modified spectral initialization scheme that outperforms prior approaches numerically. The paper is very well-written, providing optimal theoretical guarantees along with lots of interesting intuitions. The investigation of non-smooth objectives is valuable and timely for the non-convex optimization literature. Extensive numerical experiments have also been provided in comparison to prior algorithms, corroborating the practical advantages of the proposed algorithms. For these reasons, I recommend acceptance.

Qualitative Assessment

1. For the proposed spectral initialization procedure, it would be good to provide some intuition. For example, what is E[ Y ] here (with or without truncation, whatever that is simpler)? 2. It is interesting to observe that the new initialization procedure improves upon the procedure used before. Any intuition? Is it because the current version enjoys lower variance? 3. The intuition about the search direction is very nice. To make it a bit more precise, the authors might want to mention that z and {a_i} are assumed to be independent in this heuristic treatment. 4. The matching behavior between least square gradient and reshaped WF is very interesting. The authors might want to mention how they set the parameters for least square GD (I assume GD and reshaped-WF use the same step size here, right?) 5. I might have missed it but I didn't find the definition of dist(z, x) in the paper. 6. The authors might want to mention Equation (16) matches prior results (which have been shown to be minimax optimal), as those readers who are not familiar with this topic might not understand how good the bound (16) is. 7. It has been shown numerically in Figure 4 that the statistical accuracy of reshaped-WF is also better than truncated-WF. I guess this is because reshaped-WF is able to use all the data so there is no information loss, right? It would be good to provide a little explanation. 8. The paper demonstrates the power of the gradient-descent type algorithm for another (non-smooth) choice of objective function. Can the authors now say something towards a general framework on this type of algorithms? Any other objective function that can be easily accommodated by the current analysis? 9. Some minor thing: (1). Page 2: "increasing the order of a_i^T z to be four" doesn't read well. Please rewrite. (2). Page 2: "amiable to convergence" --> "amenable to convergence" ?

Confidence in this Review

2-Confident (read it all; understood it all reasonably well)


Reviewer 3

Summary

This paper studies a nonconvex formulation for the phase retrieval problem. The objective function introduced here is second order, nonconvex and nonsmooth. The authors show that in the real case, with optimal (in the order) number of measurements, gradient descent iterates of this objective linearly converges to one global optimum.

Qualitative Assessment

Comparing with previous work, including generalized phase retrieval problems, this paper has the following differences: 1) Solves a second order function incorporating absolute values of measurements 2) No step size normalization (or variants) 3) No gradient truncation The key intuition of this paper is that for a measurement |< ai, x >| with large magnitude, in the local region near global optima, the sign of < ai, x > and < ai, z > are likely to be same. In the case, locally the gradient update of reshaped WF (RWF) is as the same as an equivalent least squares problem. For the objective function including all the measurements, if most the components have reasonably large measurements, those with small measurements contribute less, hence using gradient descent + good initialization should give good results. Such intuition is formalized in equation (34) and (35). Yet I have a question -- why don’t one consider further truncating components with small magnitude when computing gradient? Just like in robust regression, TWF, etc, we can throw out “wrong” directions. This seems natural to me, according to your discussion. Though this will involve additional parameters, I am curious about it’s performance, and it might be helpful to handle the noisy case. Will this speed up or slow down convergence? Will this increase the sample complexity? One thing I would expect but missing is also the bound for the noisy case, which was covered in TWF paper. I have a few comments about the experiments: The convergence speed will be strongly affected by step sizes. For TWF, we also need to specify parameters for truncation. The authors didn’t mention how they select parameters. As the convergence rate of TWF and Reshaped WF are of the same order -- both require O(mnlog(1/eps)) flops to obtain eps-accuracy, I was expecting their behaviours are comparable. It would be more convincing if you can provide more details. Is RWF faster because the step size for competing methods are not optimal, or due to different initialization? How does their speed compare if you use the same initialization (with optimal step sizes) ? The answers would help me better understand this problem. Two more details: 1) When you introduce Algorithm 1, you haven’t define 0/0 yet. 2) In equation (36) and following proof / theorems, you should have 0.74 - 3eps, not 2eps. [1] Matrix Completion has No Spurious Local Minimum, http://arxiv.org/pdf/1605.07272v1.pdf

Confidence in this Review

2-Confident (read it all; understood it all reasonably well)


Reviewer 4

Summary

This paper proposed a new approach to solve a recovery problem expressed as the solution of a quadratic systems of equation. Compared to a recent concurrent approach, named Wirtinger flow, the proposed approach yields the minimization of a non-smooth function which behaves as quadratic in the smooth region (rather than being of order 4). The authors show that such a choice leads to improve the complexity and convergence time, hence being on a par with the truncated Wirtinger flow while being twice faster and more accurate in practice. Experimental results support the claim of the authors.

Qualitative Assessment

Since I'm not an expert in this field, I haven't really understood why the reshaped-WF can be compared to the linear problem as stated line 169-170. Figure 2 is difficult to interpret since the axis z1 and z2 are switched in (b) and (c). Hence, I did not follow how the global optimums can be compared as stated line 188. I thought a_i should be complex random measurements, but for the CDP problem in equation (17), the a_i looks to be deterministic. This is in contradiction with the forward model stated in equation (1), I believe that should have been clarified. Since the loss function in (1) is non smooth, the authors suggest using gradient descent based on Fréchet superdifferential. I was wondering whether recent proximal techniques could have been used instead since these approaches are especially developed for non smooth optimization. I believe this part of the literature might be missing in the paragraph line 94.

Confidence in this Review

1-Less confident (might not have understood significant parts)


Reviewer 5

Summary

The paper uses a nonsmooth, low-order loss function to develop an algorithm called reshaped Wirtinger flow, which improves the sample complexity of the ordinary Wirtinger flow and achieves the same sample complexity as truncated Wirtinger flow. In addition, as shown in a random Poisson noise experiment, the reshaped Wirtinger flow algorithm is more robust to noise corruption than truncated Wirtinger flow.

Qualitative Assessment

Very promising approach to solving quadratic systems of equations and have demonstrated improvement over existing algorithms. However, understanding of the reshaped WF algorithm seems to be in the infant stage. As noted in conclusion, the fast convergence is not well understood.

Confidence in this Review

1-Less confident (might not have understood significant parts)


Reviewer 6

Summary

The authors propose a method for phase retrieval which is different with WF (a previous algorithm for the same problem) in the function, and uses a non smooth one.They achieve measurement complexity nearly equal to what we have in the previously proposed method which is a truncated version of WF or TWF, and hence the sample complexity is improved in comparison with WF. Since the function is non smooth we do not have gradient here and the term gradient-like refers to what they define as gradient. The initialization is like TWF with a small change in truncation bounds which is because of removing power of two of the measurements. The gradient-like step is more close to WF since they do not use truncation in this part but they remove those points that gradient is equal to zero and claim that these points are few. There is a comparison of initialization which shows their algorithm works better. They illustrate the difference between the loss functions of their method and WF to give an intuition to reader about why their method is faster. Also there is results of the comparison between new method, WF and TWF for CDP model, Complex case and real case which shows that in complex case and CDP model the new method works better than the two others. The other experiment is for iteration count and time complexity for n=1000 and m=8n which they added Altminphase algorithm to comparison as well. For the comparison of time and iteration they also state time and iteration count in a table for image of galaxy with CDP model and L=16 and showed the recovered image with new method. Finally there is a comparison for poisson case. They proved geometric convergence and performance guarantee of the initialization and also have expectation of loss function.

Qualitative Assessment

The reviewer kindly suggests to enlarge some of the plots and to add comparison for different values of m and n, for example smaller values and also show the errors in the tables. Also suggests adding figures of other recovered images of other algorithms with the values of error and for smaller values of L to make a better comparison. The reviewer thinks it is interesting to see how the new method is working for small values of measurements or L in comparison to previous methods since L=16 is very large. It may be confusing that why the authours insist on real case while their algorithm works worse than TWF in this case and needs more measurements. If using the real case is just for simplicity of the proofs, it may be better to be explained and even just be used for the proof section. It may also cause confusion to use term gradient when there is no gradient actually and it is better to use the original word. Also it may be of great interest to add the reason behind better performance of initialization step in comparison with TWF.

Confidence in this Review

2-Confident (read it all; understood it all reasonably well)